# Tunable stochastic memristors for energy-efficient encryption and computing

Kyung Seok Woo [1,2,3,5], Janguk Han[1,5], Su-in Yi [3], Luke Thomas[4], Hyungjun Park[1], Suhas Kumar [2] ✉ & Cheol Seong Hwang [1] ✉

Information security and computing, two critical technological challenges for post-digital computation, pose opposing requirements – security (encryption) requires a source of unpredictability, while computing generally requires predictability. Each of these contrasting requirements presently necessitates distinct conventional Si-based hardware units with power-hungry overheads. This work demonstrates $Cu_{0.3}Te_{0.7}$/$HfO_2$ ('CuTeHO') ion-migration-driven memristors that satisfy the contrasting requirements. Under specific operating biases, CuTeHO memristors generate truly random and physically unclonable functions, while under other biases, they perform universal Boolean logic. Using these computing primitives, this work experimentally demonstrates a single system that performs cryptographic key generation, universal Boolean logic operations, and encryption/decryption. Circuit-based calculations reveal the energy and latency advantages of the CuTeHO memristors in these operations. This work illustrates the functional flexibility of memristors in implementing operations with varying component-level requirements.

The big data era addressed by artificial intelligence (AI) brings two critical issues—information security and the energy cost of the associated hardwares[1,2]. Memristors, or two-terminal nonvolatile memories, have been extensively studied as energy-efficient solutions for both security and computing challenges in the post-digital era of computing. Security solutions using memristors, such as the generation of true random numbers and physically unclonable function (PUF), leverage the stochastic randomness in the memristors during the switching processes[3–6]. On the other hand, memristor-based systems used for low-power in-memory computing (wherein computing and memory units are combined) strive to achieve a low degree of component-level randomness and variability[7–9]. Despite these advances, these solutions have been built using different types of memristors, often on dedicated hardware units, which limit their generality. Besides, even when memristor hardware is designed for in-memory computing, the inherently higher variability of memristors renders the hardware suitable for probabilistic computing (such as matrix

multiplications) but not for highly deterministic operations (such as arithmetic and Boolean logic). This issue further limits the generality of memristor-based computing. Thus, despite the high energy efficiency of memristor-based computing systems demonstrated for specific applications, they still fail to compete with highly flexible digital processors.

This work demonstrates a highly functional scheme of memristor-based hardware that integrates security, computing and memory capabilities using the $Cu_{0.3}Te_{0.7}$/$HfO_2$ (CuTeHO) ion-migration-driven memristor. Its stochastic switching variation is exploited as an entropy source for generating PUFs, with the added features of concealability and reconfigurability. When operated away from biases that promote stochasticity, a single CuTeHO memristor device can be guided via a sequence of up to three switching operations to exhibit all Boolean logic operations. Furthermore, this work demonstrates multiple computing processes performed on memristive hardware by exploiting the dual functional nature of the CuTeHO memristors: unclonable

[1]Department of Materials Science and Engineering and Inter-University Semiconductor Research Center, Seoul National University, Gwanak-ro 1, Daehag-dong, Gwanak-gu, Seoul, Republic of Korea. [2]Sandia National Laboratories, Livermore, CA, USA. [3]Department of Electrical and Computer Engineering, Texas A&M University, College Station, TX, USA. [4]Applied Materials Inc., Santa Clara, CA, USA. [5]These authors contributed equally: Kyung Seok Woo, Janguk Han. ✉e-mail: su1@alumni.stanford.edu; cheolsh@snu.ac.kr

key generation, logic operations, and encryption/decryption, thereby demonstrating functional flexibility that has been missing in memristor-based systems. The energy consumption and latency of these operations are notably lower compared to prevailing digital computing approaches. While there are also performance advantages (compared to prior works that used memristors) in both PUF generation and Boolean logic, the novelty of this work is the functional ability to switch between them on demand. Such flexibility enables an essential step towards functionally dense post-digital computers. As a demonstration of the utility and effectiveness of the encryption-decryption scheme, using crossbar arrays constructed using different nonvolatile memristors, matrix multiplication operations are performed on encrypted data without a noticeable loss of information.

## Results

### Influence of $Cu_xTe_{1-x}$ compositions on the resistive switching behavior

The $Cu_xTe_{1-x}HO$ memristors have different switching behaviors depending on the Cu concentration ($x$) in the $Cu_xTe_{1-x}$ electrode. It exhibits volatile threshold switching (TS) behavior at $x = 0.1$ ($Cu_{0.1}Te_{0.9}/HfO_2$, Fig. 1a, Supplementary Note 1 and Supplementary Fig. 1)[10-14], and nonvolatile resistive switching (RS) behavior at $x = 0.3$ and 0.6. Figure 1b displays the current–voltage ($I–V$) curves of the memristor with $x = 0.3$ ($Cu_{0.3}Te_{0.7}/HfO_2$, referred to henceforth as CuTeHO) memristor under a compliance current ($I_{cc}$) of 10 nA, exhibiting a self-rectifying behavior. In contrast, the $Cu_xTe_{1-x}HO$ memristor with $x = 0.6$ ($Cu_{0.6}Te_{0.4}/HfO_2$), which forms even stronger Cu conductive filaments (CFs), exhibits conductive bridge random access memory (CBRAM) behavior with Ohmic conduction in a low resistance state (LRS) (Fig. 1c and Supplementary Fig. 2a). Previous studies reported that the filament's stability is strongly influenced by its surface curvature[10-14], which determines the device to be either a diffusive memristor or a CBRAM. However, at $x = 0.3$, the device exhibits nonvolatile and rectifying behavior, which deviates from the typical characteristics of CBRAM. It exhibits an Ohmic conduction property immediately after being switched to the LRS by the $I–V$ sweep (Curve 1 in Fig. 1b). However, the subsequent sweep does not exhibit Ohmic conduction, suggesting the potential coexistence of another switching mechanism (Curve 2). The conduction behavior observed in the LRS and the rectification characteristics in the negative bias region

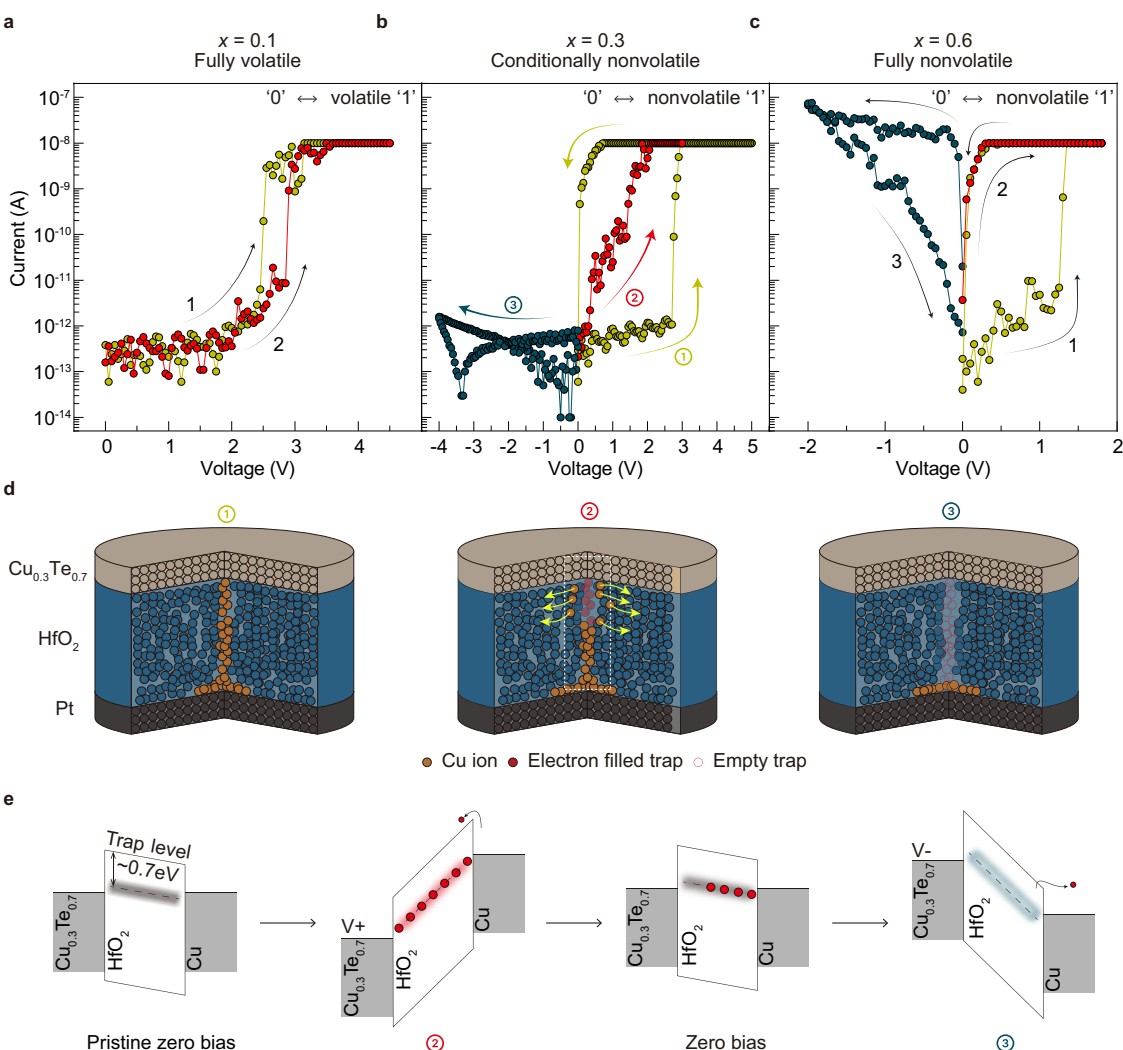

**Fig. 1 | Influence of $Cu_xTe_{1-x}$ electrode compositions on the resistive switching behavior.** $I–V$ curves of the $Cu_xTe_{1-x}/HfO_2/Pt$ device at **a** $x = 0.1$, **b** $x = 0.3$ and **c** $x = 0.6$. A compliance current ($I_{cc}$) of 10 nA is set to both devices during the positive voltage sweep. **d** Switching mechanism of $Cu_{0.3}Te_{0.7}/HfO_2/Pt$ device corresponding to (**b**). **e** Schematic band diagram of the $Cu_{0.3}Te_{0.7}/HfO_2/Pt$ device after the filament formation in the dotted box region in (**d**). There is a small internal electric field by the work function difference between two electrodes. When the positive voltage is applied to the $Cu_{0.3}Te_{0.7}$ electrode, the trap level is lowered below the Fermi energy level of the Cu electrode, and the electrons injected from the Cu electrode fill the traps. Once the traps are filled with electrons, the electrons cannot be detrapped at zero bias due to the weak built-in potential, maintaining the nonvolatile behavior.

(Curve 3) are similar to that of an electronic switching memristor, where electron trapping and detrapping occur at a trap layer[15–18]. In addition, the $I–V$ characteristics of the CuTeHO memristor were examined at a higher $I_{cc}$ of 100 nA (Supplementary Fig. 2b), where sufficiently strong CFs form, which cannot be ruptured at zero bias. Similar to the $Cu_{0.6}Te_{0.4}/HfO_2$ device, no rectifying behavior was found due to the stronger CF at higher $I_{cc}$. Furthermore, the $I–V$ curves of the LRS device were measured at different temperatures (Supplementary Fig. 3), where the absence of a decrease in the current in the LRS device with increasing temperature indicates that the conduction is not solely based on a metallic CF. Based on the varying qualitative memristive behaviors, the switching mechanism of the CuTeHO memristor can be understood as a combination of CF formation/rupture and electron trapping/detrapping, as illustrated in Fig. 1d. The relatively weak Cu CFs are formed during the SET (switching from high resistance state (HRS) to LRS) process. At zero bias, the diffusion of the Cu atoms from the CF to the surrounding $HfO_2$ region causes a slight rupture of the filament (similar to the behavior observed in the $Cu_{0.1}Te_{0.9}/HfO_2$ device), simultaneously creating a localized path for charge trapping. In other words, the residual Cu CFs near the bottom electrode (BE) can be considered an extended electrode. The electron trapping/detrapping process in the local CF-ruptured region mediated by traps at roughly 0.7 eV below the conduction band (Fig. 1e), reported elsewhere[18], can explain the observed switching behaviors. Kim et al.[19] confirmed this switching mechanism, where the insulator was divided into switching and filament regions. The self-rectifying behavior in the negative bias region is attributed to the slightly ruptured filament and a possible barrier formed at $Cu_xTe_{1-x}/HfO_2$ interface due to the Fermi level pinning.

The self-rectification of the CuTeHO memristor can effectively suppress sneak currents when integrated into a passive memristive crossbar array, making it highly attractive for applications requiring low power consumption and high density. To further validate the practical use of the device in a large-scale array, using circuit-based simulations, sneak currents and line resistance issues were evaluated as the array size was increased (Supplementary Note 2). With the sneak current suppressed by the rectifying behavior, no critical issues were observed in array sizes up to $400 \times 400$ in the worst-case scenario and $1000 \times 1000$ when half the devices were at HRS (Supplementary Fig. 4). The line resistance did not affect the operation in a large-scale array (Supplementary Fig. 5). Additional details about the device are provided in Supplementary Figs. 6 and 7. The volatile and nonvolatile behaviors for the cases of $x = 0.1$ and $x = 0.6$ are not surprising as reported elsewhere[20–23]. However, the dual behavior at an intermediate stoichiometry ($x = 0.3$) is unexplored and is utilized for diverse applications in this work, which are explained in the following sections.

## Physically unclonable functions with concealability and reconfigurability

PUF is a hardware security primitive that utilizes the inherent variations arising from individual components, such as manufacturing variations, cycle-to-cycle noise, and chaos, thereby providing each hardware with distinct properties and fingerprints[24]. PUF can be used for identity authentication and generating cryptographic keys for encrypting confidential or private data. The encryption process using the CuTeHO memristor is described in more detail later. Each PUF is associated with a unique output ('response') mapping from an input ('challenge'), forming a challenge-response pair. Various PUF systems have been reported based on optical components, arbiters, static random-access memory, and ring oscillators[20–23]. However, the previously reported PUFs faced limitations with regard to scalability, fabrication complexity, non-trivial operation, and post-processing issues.

In contrast, the intrinsic stochasticity of memristors during their switching process holds great promise for hardware security

applications owing to their high switching speed, low power consumption, and remarkable potential for scalability[25–30]. A critical requirement for modern PUFs is reconfigurability, where keys (the challenge-response pairs) are regenerated within the same chip to account for the increasing probability of information leakage with time. A second critical requirement is the concealability of the key when stored in hardware. There have been reports on reconfigurable memristive PUFs[26–29], for instance, using conductance variations of neighboring cells to achieve reconfigurability. However, these reconfigurable memristive PUFs were not area-efficient as they required at least two memristors to generate a single bit. Furthermore, there has been a report of the concealment of keys in a memristive PUF, where the data was concealed by switching all devices into the LRS[30]. However, no memristive PUF system has offered both concealability and reconfigurability.

This work introduces a novel memristive PUF hardware that is concealable and reconfigurable, using the stochastic variations in the SET voltage of the CuTeHO memristor. A distribution of SET voltages from 400 cycles in 8 memristors was obtained using the cycle-to-cycle and device-to-device variations, with a median bias of 2.85 V used as the challenge that applies to all memristors for generating the PUF (Fig. 2a). The bit response is represented as '1' when the memristor switches to the LRS and '0' when it remains in HRS. Thus, this PUF hardware is area-efficient as each memristor represents one bit.

The experimentally fabricated $4 \times 4$ crossbar composed of the CuTeHO memristors was employed to demonstrate the memristive PUFs (Fig. 2a, b), of which fabrication procedure is provided in Supplementary Fig. 8. Concealing-revealing and reconfiguration processes are illustrated in Fig. 2c, d. The median value (2.85 V) was initially applied to all the memristors. This operation rendered half of them switched to LRS and the other left at HRS, due to the stochasticity of the CuTeHO memristors, generating PUF-1 (or base PUF) displayed in Fig. 2b. Subsequently, an additional $I–V$ sweep down to −3.5 V, which partially ruptures the CFs in the LRS cell (step 2 of Fig. 2c), was applied to all the devices. This operation concealed the original PUF data because all the cells switched to HRS (step 3 of Fig. 2c). During concealment, the CFs rupture to different extents depending on the original state (LRS ruptures partially, while HRS ruptures fully; see Supplementary Fig. 9). Upon applying +2.85 V, the partially ruptured CFs were restored to the LRS, while the fully ruptured ones remained in the HRS, thus revealing the original key (step 4 of Fig. 2c). In this way, PUF-1 was concealed and revealed. When a key is discarded and a new key is desired (reconfiguration), all the memristors are made to undergo full SET (3.5 V) and full RESET (−4.0 V) cycles, which are followed by another $I–V$ sweep to 2.85 V, as displayed in Fig. 2d. Once the device is programmed to its HRS, the SET voltage randomly changes after the RESET process (Supplementary Fig. 10). In other words, there is no history-dependent behavior that can compromise the randomness of a newly generated key. Hence, the PUF can be reconfigured with a different pattern of LRS/HRS cells due to the inherent stochasticity.

By including two consecutive reconfigurations of PUF-1, 20 different $4 \times 4$ PUFs were produced (the $16 \times 20$ bitmap in Fig. 2e and Supplementary Fig. 11). From these PUFs, the inter-Hamming distance (inter-HD) was calculated by comparing the bit difference between two PUFs. The histogram of the inter-HD was fitted with a Gaussian distribution with a mean value of 0.5063, showing excellent uniqueness of the memristive PUFs (Fig. 2f). In addition, all the collected bits in this work passed the National Institute of Standards and Technology (NIST) randomness test (Supplementary Table 1). Moreover, the conceal-reveal performance was examined through 10 consecutive cycles starting at PUF-1, achieving bit error rates of 1.88% for concealing and 0% for revealing (Fig. 2g and Supplementary Fig. 12).

Table 1 provides a comparison with other memristive PUFs. Among the qualitative features of reconfigurability, concealability, nonvolatility and logic capability, all prior efforts were unable to

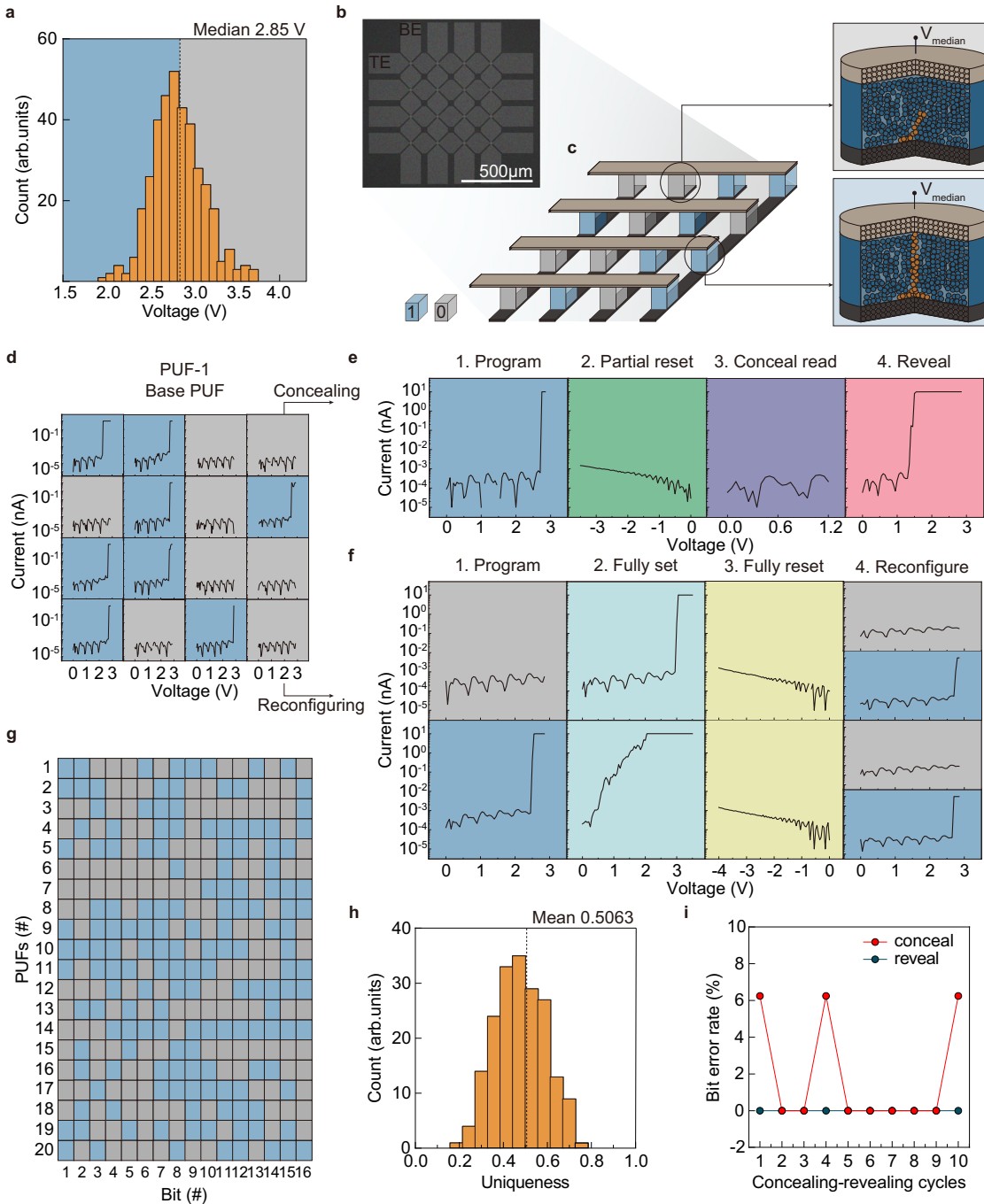

**Fig. 2 | Memristor-based PUF design. a** Distribution of SET voltages from 8 memristors. A PUF hardware is based on a memristive crossbar array, where each memristor is represented as a PUF bit depending on the filament formation at $V_{median}$. **b** A scanning electron microscope image of the CuTeHO memristive crossbar used in the experiments. **c** An illustration of the crossbar array with memristors programmed to represent a 1 or a 0. Illustrations of two states of the memristor are also provided. **d** 4 × 4 memristive PUF 1. **e** Concealment process at each stage. The partial RESET voltage is applied to all the memristors to switch them into partial HRS to conceal the PUF data. The memristors, which were previously in LRS, will have a lower SET voltage, and only those devices will return to LRS when the challenge is applied, thereby revealing the concealed PUF data. **f** Two examples of reconfiguration with different initial responses ('0' and '1'). The first reconfiguration step involves applying a sufficiently high voltage (SET voltage) to ensure that all memristors are in the LRS. Afterward, the memristors are assigned a new SET voltage through the RESET process. **g** 16 × 20 bitmap generated from 18 different memristive crossbars. **h** Inter-Hamming distance (inter-HD) of the PUFs, fitted with a Gaussian distribution. **i** Bit error rate of concealing-revealing cycles.

demonstrate at least two of them. This work provides all four qualitative features, in addition to area efficiency (a single cell represents a full bit). The device's self-rectifying behavior also enables its implementation in selectorless (passive) crossbar array configuration, whereas prior works required complex design using transistors or selectors in order to be included in scalable crossbar arrays.

The logic capabilities of the CuTeHO memristors are elaborated in the following section.

## Dual conditional logic operations
Various methods of memristor-based in-memory computing have been proposed that can be categorized into stateful and nonstateful

**Table 1 | Comparison of memristor-based PUFs**

| | This work | Zhang et al.[25] | Nili et al.[26] | Jiang et al.[27] | John et al.[28] | Gao et al.[30] |
|---|---|---|---|---|---|---|
| Device structure | Simple ($Cu_xTe_{1-x}$/ $HfO_2$/Pt) | Simple (Au/Ag:$SiO_2$/Au) | Complex (Pt/TiN/$TiO_{2-x}$/ $Al_2O_3$/Pt/TiN/ $TiO_{2-x}$/$Al_2O_3$/ Pt/$TiO_2$/$Al_2O_3$) | Simple (Ta/$HfO_2$/Pt) | Complex (Ag/PMMA/ PrPyr[PbI3]/ PEDOT:PSS /ITO) | Simple (TiN/$TaO_x$/ $HfO_x$/TiN) |
| Bits per cell | 1 | 1 | 1/ # of rows | 0.5 | 0.5 | 1 (needs extra current source) |
| PUF methodology | SET voltage variations | Device switchability based on Ag concentrations | I–V nonlinearity variations | LRS of neighboring cells | HRS of neighboring cells | HRS compared to a reference |
| Uniqueness (%) | 50.62 | 50.68 | 49.96 | 50.06 | 49.02 | 50.52 |
| NIST test | Passed | Not reported | Passed | Not reported | Passed | Passed |
| Reconfigurability | Yes | No | Yes | Yes | Yes | No |
| Concealability | Yes | No | No | No | No | Yes |
| Nonvolatility | Yes | No | Yes | Yes | Yes | Yes |
| Logic operation capability | Yes | No | No | No | No | No |
| Implementable with passive nonvolatile crossbar | Yes | No | No | No | No | No |

logic. In stateful logic, both logic input and output are represented by an identical information carrier, such as a memristor's resistance[7,31–34]. On the other hand, nonstateful logic adopts different information carriers, such as voltage input and resistance output[35–38]. Stateful logic usually requires more memristors than nonstateful logic, but the logic cascading is more straightforward. In contrast, reconfiguring the nonstateful logic is more accessible than reconfiguring stateful logic since the logic operation can be adjusted by varying the input voltages. Because the CuTeHO memristor in this work possesses reconfigurability, a nonstateful logic approach is employed, and all 16 basic Boolean logic operations are demonstrated using a single device. This logic scheme has two voltage conditions, termed dual conditional logic (DC logic).

The fundamental DC logic operation method using the CuTeHO memristor is presented in Fig. 3a. It consists of logic inputs ($TB$), where the input voltage is applied to top ($T$) and bottom ($B$) electrodes, and the resulting resistance state of the memristor represents the logic output $Z$. The DC logic concept using a single CuTeHO memristor can be understood by a finite state machine (FSM), as illustrated by Fig. 3b. Condition 1 corresponds to the SET process (V = $|V_{set}|$, $I_{cc}$ = 60 nA). In this condition, the $I_{cc}$ of 60 nA slightly increases the RESET voltage of the CuTeHO memristor due to a thicker filament formation, which prevents the partial RESET at −3.5 V (Fig. 3c and Supplementary Fig. 13a). When the initial resistance state $Z'$ is '0', only the logic inputs $TB$ of '10' can switch the device into LRS ('1'). This operation corresponds to $T$ NIMP $B$, equivalent to $T$ AND(NOT$B$). However, $Z'$ does not change at any inputs when it is '1'. Both cases should be considered together in condition 1. Furthermore, condition 2 corresponds to the RESET process with a different voltage and $I_{cc}$ (V = $|V_{reset}|$, $I_{reset}$ < $I_{cc}$ < $I_{set}$). When a positive $V_{reset}$ is applied with an $I_{cc}$ of 50 pA, the CuTeHO memristor cannot fully switch from HRS to LRS, resulting in TS behavior due to weak filament formation at this low $I_{cc}$ (Supplementary Fig. 13b). On the other hand, a negative $V_{reset}$ enables switching from LRS to HRS regardless of $I_{cc}$ level, due to the self-rectifying characteristics of the CuTeHO memristor. When $Z'$ is '1', only the logic inputs $TB$ of '01' can switch the device to HRS ('0'), thus executing $B$ IMP $T$, which is equivalent to (NOT$B$)OR$T$. To summarize, the two voltage conditions can be defined as follows:

$$Condition\ 1 : Z = T \cdot \bar{B} \cdot \bar{Z}' + Z' = T \cdot \bar{B} + Z' \quad (1)$$

$$Condition\ 2 : Z = (T + \bar{B}) \cdot Z' \quad (2)$$

The experimentally measured state transition processes of the FSM are displayed in Fig. 3d. All 16 logic operations are experimentally demonstrated by sequentially applying voltage conditions specific to each logic operation in no more than three steps to substantiate the viability of the suggested concept further (Fig. 3e and Supplementary Figs. 14–19). The logic operations can also be performed in a pulse mode by controlling the pulse amplitude and width. The pulse operation of the CuTeHO memristor is provided in Supplementary Fig. 20. Compared to other related studies that conducted logic operations with a single device, the CuTeHO memristor offers advantages of fabrication simplicity and operational efficiency over the one diode-one resistor (1D1R) approach[39] or the trilayer-oxide-based unipolar resistive switching (URS) device approach[40]. Using the suggested DC logic method, a 1-bit binary full adder/subtractor was constructed using a 2 × 2 crossbar in 8 steps (Supplementary Notes 3 and 4, Supplementary Figs. 21–23). Utilizing the structural and functional characteristics of the crossbar, this work shows that the number of required devices and operation steps are lower than previously reported memristor-based adders (Supplementary Table 2). Especially, the suggested method has a significantly lower energy cost than alternatives. Furthermore, the adder and subtractor can be combined in a single 3 × 2 memristive crossbar, allowing concurrent mapping of outputs for both operations (Supplementary Note 4.3, Supplementary Figs. 24–27 and Supplementary Table 3). These proposed DC logic-based transistorless schemes substantially reduce the area and computational costs of in-memory logic operation while enabling enhanced functional density in a single component.

## Encryption and decryption of data

Among the 16 logic operations demonstrated in the previous section, an XOR logic operation can effectively encrypt and decrypt data using cryptographic keys[41]. For instance, the XOR logic operation of the 'A'-shaped data and cryptographic key converts the original data into encrypted data, which can then be decrypted by another XOR logic operation with the same key (Fig. 4a). Since the CuTeHO device can generate keys using its SET voltage variation, it is possible to encrypt and decrypt data by combining the key generation with 3-step XOR DC logic, both using the CuTeHO devices.

Here, 20 binary letter-shaped data represented by a 4 × 4 pixel grid were flattened into 1 × 16 sized vectors and mapped onto a 20 × 16 memristor array (Fig. 4b). In this case, the applied SET voltage should be higher than 3.5 V to mitigate the impact of SET voltage variation (since the target is an information storage and not PUF generation).

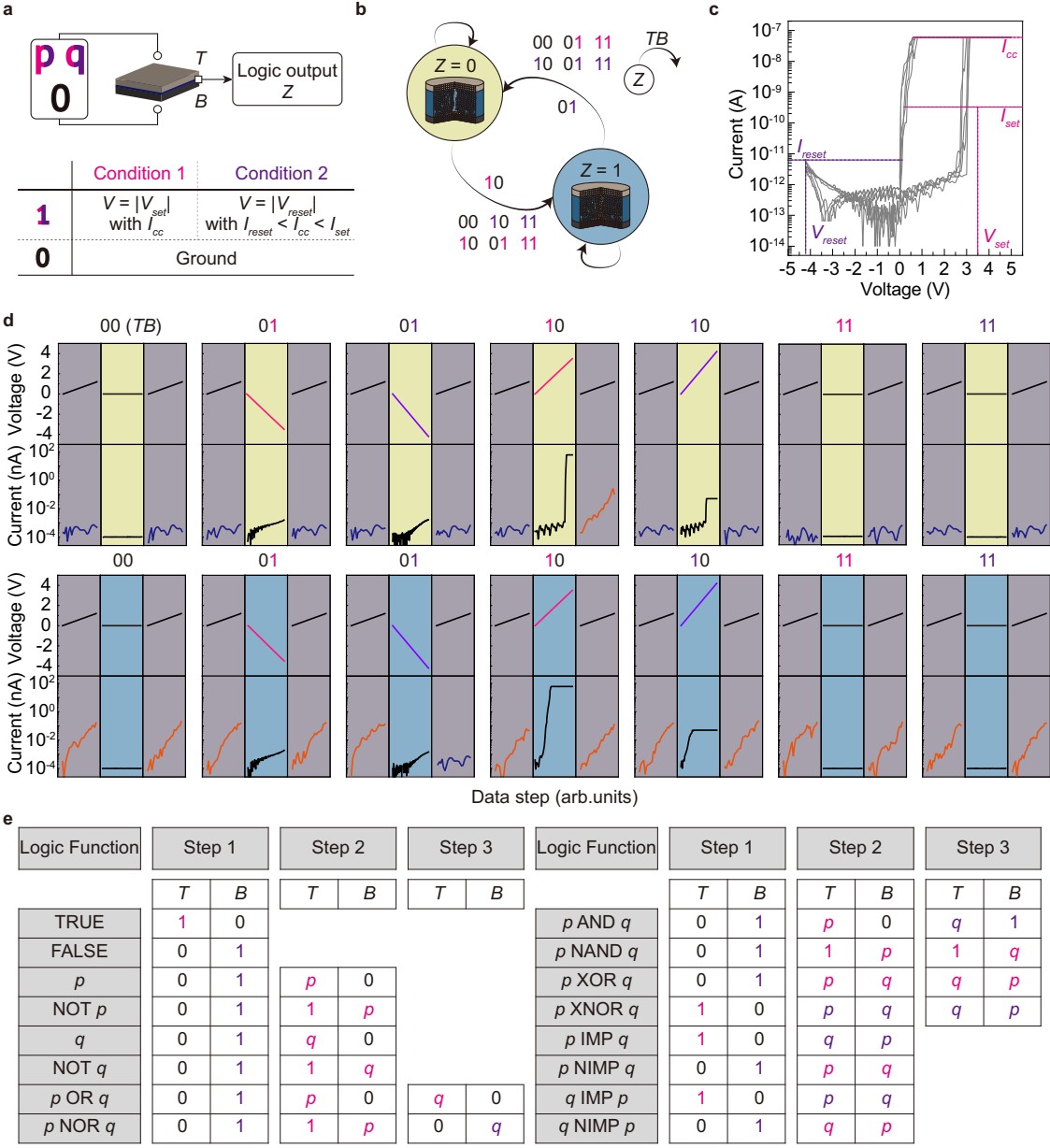

**Fig. 3 | Dual Conditional (DC) logic scheme based on the CuTeHO device.**
**a** Schematic diagram and two conditions of the DC logic. Logic input '0' is defined as the ground potential, and logic input '1' is defined as one of the two conditions. Similar to the memristive PUFs, the LRS and HRS of the memristor are represented as logic outputs '1' and '0', respectively. **b** Finite state machine of the CuTeHO device. **c** I–V curves of the CuTeHO device at $I_{cc} = 60$ nA. **d** Experimental demonstration of the finite state machine. Two gray panels in each subfigure are read operations before and after the logic operation. Yellow and blue panels show the transition processes when the initial resistance state is '0' and '1', respectively. **e** Logic sequences for 16 logic operations.

The self-rectifying characteristic of the CuTeHO memristor effectively suppresses leakage current when data is mapped. In another 20 × 16 sub-array, which matches the size of the memory array, a set of PUF keys was generated by setting the voltage of each memristor to 2.85 V. These keys can be reconfigured or concealed as needed. Overall, a 20 × 32 hybrid memory-PUF array was created, demonstrating the dual nature of storing deterministic quantities and generating unpredictable quantities.

A 1 × 16 shaped data and a 1 × 16 PUF key were used as inputs to the word lines (WLs) and bit lines (BL) of a 16 × 16 logic operation array to encrypt the selected shape data using the XOR DC logic scheme (Supplementary Fig. 28a). When the voltages are applied to WLs and BLs in the 16 × 16 encryption array, only the cells at the main diagonal experience the appropriate voltage for the XOR operation of the encryption. Therefore, the result of the logic operation is mapped to

the main diagonal. After the encryption, the information on the main diagonal was read, replacing the original letter-shaped data. The PUF array can also be concealed via partial RESET, ensuring that the original data and cryptographic key remain hidden until the decryption process (demonstrated in Supplementary Fig. 28b). The overall results of the encryption using 20 different 16-bit letter-shaped data ('A' to 'T') with 16-bit cryptographic keys (PUF-1 to PUF-20) are displayed in Supplementary Fig. 29. The data and cryptographic keys can be mapped through a microcontroller-based circuit, and the encryption can also be controlled using the same circuit (Supplementary Fig. 30 and Supplementary Note 5). Furthermore, it was examined if mathematical operations can be performed on encrypted data in noise-prone analog computers, without losing the original information (Supplementary Notes 6 and 7). The experiment using analog memristor chips to perform AI inference on encrypted data confirmed that there was

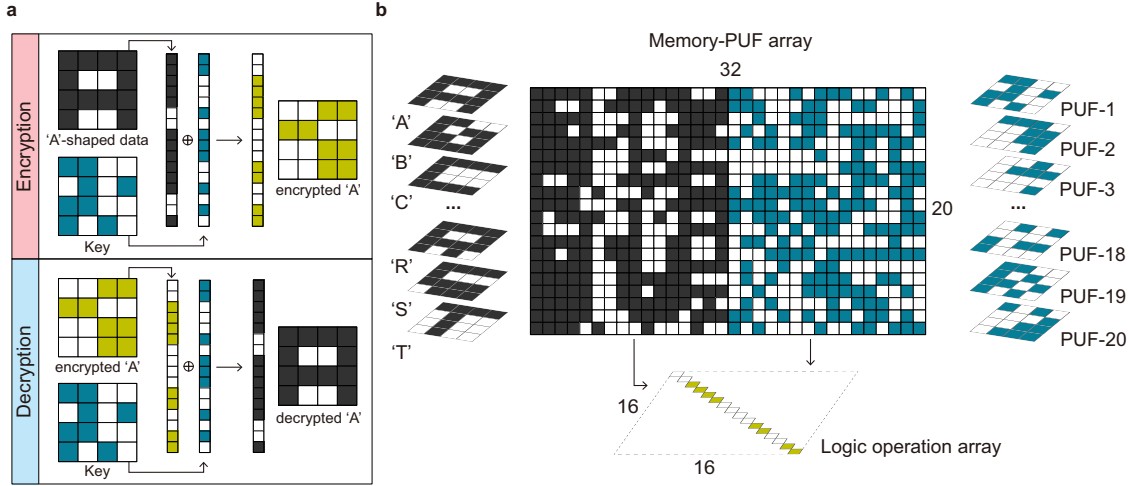

**Fig. 4 | Hardware encryption/decryption based on CuTeHO memristive crossbar. a** Encryption/decryption of an A-shaped data. The key is equivalent to PUF 1, while the encryption/decryption is performed through an XOR operation based on the DC logic scheme. **b** 20 × 32 hybrid mem-PUF array utilized for encryption.

minimal degradation of the overall quality of the inference compared to inference performed on unencrypted data. Thus, the encryption-decryption scheme is robust to noise and minor variabilities typical of analog computers.

Table 2 compares the performance between the memristor-based approach in this work (using projected calculations) and a CMOS-based computer. The energy consumption and latency are lower in the memristor-based approach, especially the energy needed to perform encryption/decryption using XOR logic (details provided in Supplementary Note 8, Supplementary Figs. 35 and 36). These comparisons are based on published projections, which may take a few more years to reach volume production, by when CMOS computers would also be improved. These comparisons indicate the competitiveness of memristor-based technologies, while the quantitative estimates may evolve with technological progress.

## Discussion

CMOS-based logic and computing have been highly reliable for decades and have employed arbitrary reconfigurability as the backbone of general-purpose computing. However, performance of digital computers is generally saturating[42], leading to increased costs for slower improvements in performance. As such, there is a clearly identified need for post-CMOS technologies[2,39], which generally promise better performance (for instance, in terms of energy consumption) compared to CMOS approaches. However, a key issue with post-CMOS technologies is their application specificity due to various fundamental reasons, such as their frequent reliance on hard-coded physical processes to perform computing. Such specificity implies the need for

discrete accelerating units for different tasks (such as logic, matrix operations, etc.). Therefore, the next wave of research into post-CMOS technologies will need to address functional reconfigurability and application flexibility.

The CuTeHO memristors offer the first steps towards functionally reconfigurable post-CMOS components. These memristors embody fundamentally tunable ion-migration-based kinetics, which are reflected as various device-level functions with opposing requirements. The CuTeHO-based PUF-generating memristors exhibit various such functional features: reconfigurability, concealability, nonvolatility, logic capability, and scalability via passive crossbar arrays. Thus, the CuTeHO memristors not only offer improved performance, but promise functional generality as well. Such application flexibility, especially of post-CMOS devices that can express universal Boolean logic, is a well-recognized need[40].

While the performance estimates are based on experimentally calibrated projections and not direct experimental comparisons to entrenched CMOS technologies at full chip scales, these promising projections (among other memristor-based technologies) will aid the down-selection of post-CMOS technologies that will be moved to large-scale production. Many similar comparisons to other memristor technologies will serve this purpose[6,28,43]. Such large-scale production of post-CMOS technologies is anticipated to be realized within the next decade[2,39].

In summary, this work experimentally demonstrated a compact and multifunctional hardware system in a CuTeHO memristive crossbar architecture that can be used for probabilistic (key generation, encryption/decryption) and deterministic (logic operation) computations, which often have opposing requirements−the former requiring unpredictability and the latter requiring predictability. The PUF, which serves as a digital fingerprint or key generation, relies on the stochastic switching behavior of the memristor. In addition, nonstateful logic was implemented based on binary states induced by different biases. These functions are used to demonstrate data encryption and decryption. Performance calculations show a clear advantage in terms of both energy and latency in the demonstrated encryption operation. This work paves the path for multifunctional memristive systems with an increased spectrum of applications.

## Methods
### Device fabrication
The 4 × 4 memristive crossbar array consisting of $Cu_xTe_{1-x}/HfO_2/Pt$ devices was fabricated with the following procedure (Supplementary Fig. 8). A 10-nm-thick Ti adhesion layer and a 50-nm-thick Pt bottom

**Table 2 | Energy and latency comparisons of XOR encryption (encrypting 'A'-shaped data with PUF-1, as shown in Supplementary Fig. 29) between the memristor-based approach and CMOS-based computers**

|  | Memristor | | CMOS computer | |
| --- | --- | --- | --- | --- |
|  | **Energy** | **Latency** | **Energy** | **Latency** |
| RNG | 34.4 pJ | 23.5 ns | 144 pJ | 94 ns |
| Encryption/decryption | 99.9 pJ | 54.4 ns | 288 pJ | 0.14 ns |
| SUM | 134.3 pJ | 77.9 ns | 432 pJ | 94.1 ns |

The calculations are based on the hardware setup described in Supplementary Figs. 30 and 35, and account for the peripheral support circuits in addition to the memristor circuits.
For RNG, see Supplementary Note 8.1 and Figure S35. For encryption/decryption, see Figure S35 for memristor calculations, and Supplementary Note 8.2 (and Figure S36) for CMOS computer.

electrode (BE) were deposited on a $SiO_2$/Si substrate using an electron beam evaporator (SRN-200, SORONA), followed by a lift-off process. A 10-nm-thick $HfO_2$ switching layer was deposited via thermal atomic layer deposition using $Hf[N(CH_3)(C_2H_5)]_4$ and $O_3$ as Hf precursor and oxygen source, respectively. A 50-nm-thick $Cu_{0.3}Te_{0.7}$ top electrode (TE) was DC sputtered by co-sputtering from Cu and Te targets (07SN014, SNTEK) with a power of 20 W and 55 W, respectively. The power and time of co-sputtering can be adjusted to achieve the desired x value for other $Cu_xTe_{1-x}$ devices (Supplementary Table 4). Finally, a 40-nm-thick Pt passivation layer was deposited using the electron beam evaporator, followed by the lift-off process. An additional dry-etching system (PlasmaPro System 100 Cobra, Oxford) was used to make the BE contact for the crossbar fabrication.

## Electrical measurements

The DC current–voltage ($I$–$V$) characteristics of the devices were measured using the semiconductor parameter analyzer (SPA, HP4155B). During the measurement, the TE was biased, and the BE was grounded. The AC pulse measurement was performed using the SPA, pulse generator (PG, Agilent 81110A), and oscilloscope (OSC, Tektronix TDS 684C). An interface built on LabVIEW was used for all electrical measurements.

## Data availability

All the relevant data are available from the corresponding authors upon request.

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

## Acknowledgements

R. Stanley Williams is gratefully acknowledged for feedback on the experimental results and assistance with the device analysis. A part of this work was supported as part of the Center for Reconfigurable Electronic Materials Inspired by Nonlinear Neuron Dynamics (reMIND), an Energy Frontier Research Center funded by the US Department of Energy (DOE), Office of Science, Basic Energy Sciences. Specifically, reMIND supported efforts from K.S.W. and S.K. on theoretical modeling, simulations, performance calculations, electrical measurements (K.S.W.), and project co-supervision (S.K.). Any subjective views or opinions that might be expressed in the paper do not necessarily represent the views of the US DOE or the United States Government. Sandia National Laboratories is operated for the US DOE's National Nuclear Security Administration under contract DE-NA0003525. This work was also supported by the National Research Foundation of Korea (Grant No. 2020R1A3B2079882).

## Author contributions

K.S.W. designed the study concept. K.S.W. and J.H. performed electrical measurements. J.H. fabricated the device and refined the concept. S.Y. performed energy and latency calculations. L.T. oversaw the design, assembly and operation of the hardware for AI inference. H.P. contributed to the device analysis. S.K. suggested the encrypted inference concept and performed related experiments and modeling. S.K. and C.S.H. supervised the entire project. K.S.W., J.H., S.K. and C.S.H. wrote the manuscript.

## Competing interests

The authors declare no competing interests.
