## [Peer Review File · Nature Communications]

REVIEWER COMMENTS

Reviewer #1 (Remarks to the Author):

In Tunable Stochastic Memristors for Energy-Efficient Encryption and Computing, Woo et al. present CuTeHO based-memristor hardware for various functionalities including physically unclonable functions, logic operations, random number generation, encryption/decryption and other useful functions.

The manuscript is written very ambitiously with statements such as "Thus, memristor-based hardware, [...] have matured beyond efficient implementations of isolated functions and can be used for an end-to-end solution combining computing and security"

There is a 62-page supplementary with pages after pages of results.

Unfortunately, when examined carefully, I don't think the manuscript delivers its promises or demonstrates what it says it demonstrates. The supplementary information is put together in a manner that makes it extremely hard to read or evaluate, with many details missing.

Let us take the application of PUF. Despite what the authors intend with "maturity", one of the first obvious applications of any new nanotechnology is PUFs, since, by definition, the variations and uncontrollability in device fabrication are used as a function (making lemonade out of lemons). So PUF does not seem to be the best application for a mature technology. Even then, the experimental results are based on 4x4 crossbars.

Unlike what the authors claim, this is still at the level of laboratory experiments, with some basic demonstration. PUFs with memristors of course is not new, as the authors are aware, similar approaches, for example in Ref. 26 have implemented security primitives with PUFs.

Here, the authors make some claims about exposed keys and reconfigurability, but even if these are accurate, it is unclear why a 4x4 PUF with some reconfigurability is such a major advance or why it points to maturity. In many cases with PUFs, it is unclear to me why one would need to refresh the key so much, or why these prior approaches could not refresh their keys (not explained well), or why this is so important to advance the field.

Next, the authors move on to some basic logic gates with new devices. Once again, much of their experimental results are extremely basic (and hence I find their GPU/CPU comparisons disingenuous). These functions are performed with hundreds of billions of transistors in CPUs and GPUs whereas in FIG. 3 we are observing some basic finite state machine (FSM) with a handful of states (less than 10) and somehow we are to believe in the maturity of the sequential logic capabilities of the memristive approach.

The device community has largely moved on from trying to implement basic Boolean logic with new devices because CMOS transistors are impossible to beat in this space. The authors can check Dmitri Nikonov's work on benchmarking Boolean logic with a variety of new technologies. Unfortunately, the

implementation of a 4x4 PUF or a 4-state FSM do not seem like the "end-to-end" solution this paper claims.

What follows is a basic encryption/decryption demo with slightly larger arrays of memristors preceding a Memristor / CPU / GPU table comparison.

I found it hard to follow how this table is obtained, since the numbers are well-hidden in a disorganized 62-page supplementary.

Take RNG of CPU for example. The paper the authors cite from Intel can generate random numbers with a throughput of 323 Gbit per second. In any real application that needs RNGs, requiring trillions and trillions of random numbers are very common. What is the throughput of the memristor approach that leads to 51.7 pJ of energy consumption? How are we sure that the size, throughput and statistical properties of RNGs are equivalent? I see some NIST test comparisons, but I am curious whether the authors really believe that their approach on all of these "training, RNG, Inference, Encryption" functions can really outperform CPUs and GPUs, because, as they would appreciate, winning against CPUs or GPUs in ONE of these applications convincingly would be a win for the field.

Overall, I find the paper highly overstating what it actually delivers. Unfortunately the comparisons seem unconvincing and superficial. A much more tempered manuscript could deserve publication, but I am not sure if in Nature Communications.

Reviewer #2 (Remarks to the Author):

Author(s) presented good work in the domain of encryption and computing simultaneously with CuTeHO based stochastic memristors. The whole manuscript is well organized and useful results are demonstrated. This work can be considered but few points need to be addressed.

1. Clearly explain the variability of molar fraction 'x' in the development of PUF and computing system.
2. Give more clarity of deep learning model used for encryption of data.

Reviewer #3 (Remarks to the Author):

This paper's biggest claim is that they have created a novel memristor that they can use both for computing and security purposes. The authors claim that under specific biases, they can either induce the stochastic behavior required to have a memristor be able to generate truly random numbers (TRNG) and unclonable functions (PUFs) or perform universal Boolean logic. The trick is to tune the strength of the conductive filament, which stays weak (volatile) when the current is under 10 nA and non-volatile if the current is above 100 nA.

The device consists of a 50 nm Cu/Te top electrode, a 10 nm HfO for their switching layer, and a 50 nm Pt for the bottom electrode. The authors settled on a 30% concentration of Cu to Te for the top electrode as this device has non-volatile characteristics and self-rectifying behavior. The operating range for the device seems to be -3.5 V for reset and 2.5 V for set (they have shown extensive statistical data to arrive

at those values). A weak conductive filament formed when the compliance current I_{cc} was set to 10 nA, which could be easily broken with a slight reset. Those fractions from the broken conductive filament serve as traps to induce the stochastic behavior necessary for PUFs.

When the device is operated at 100 nA I_{cc} , this stochastic behavior doesn't exist as the conduction filament is much stronger and more challenging to break. The authors experimentally demonstrated dual conditional logic operations and encryption and decryption.

The authors had extensive data to support most of the claims. The supplemental file is filled with stability, cycle-to-cycle, pulse behavior, and retention data on the device. Their claims were all backed up with experimental data, not simulated.

However, there are some critical issues that should be addressed. First, the device physics component is relatively weak, with only a handwaving explanation. Some level of physical characterization and simulation are necessary to strengthen this part. Second, the sneak path issue will be a showstopper for scaling up the CuTeHO devices technology. It is suggested that the authors provide an analysis of how large an array can be operational for logical operations. Third, the benchmarking is not specific to certain models of GPU and CPU, and it is vague if the external PCB boards are included in the benchmarking. Finally, the 1T1R array part is disjointed from the main idea of the manuscript. It is not the CuTeHO device, but an existing cell technology from Applied Materials. Similar type inference engines (with much larger scale) have been demonstrated in both academia and industry. It is an unnecessary decoration, and I suggest the authors remove this part of irrelevant data.

Jan 31, 2024

Our responses to the comments are provided below. We also supplied revised main text and supplement, marked with red-colored text to indicate revisions.

We welcome any additional comments that the reviewers may have.

Black (italics): Reviewer comments

Blue: Author responses

Red: Modified text in the manuscript/supplement.

Black (non-italic): original text in the manuscript/supplement.

Reviewer #1:

In Tunable Stochastic Memristors for Energy-Efficient Encryption and Computing, Woo et al. present CuTeHO based-memristor hardware for various functionalities including physically unclonable functions, logic operations, random number generation, encryption/decryption and other useful functions.

The manuscript is written very ambitiously with statements such as "Thus, memristor-based hardware, [...] have matured beyond efficient implementations of isolated functions and can be used for an end-to-end solution combining computing and security"

There is a 62-page supplementary with pages after pages of results.

Unfortunately, when examined carefully, I don't think the manuscript delivers its promises or demonstrates what it says it demonstrates. The supplementary information is put together in a manner that makes it extremely hard to read or evaluate, with many details missing.

Let us take the application of PUF. Despite what the authors intend with "maturity", one of the first obvious applications of any new nanotechnology is PUFs, since, by definition, the variations and uncontrollability in device fabrication are used as a function (making lemonade out of lemons). So PUF does not seem to be the best application for a mature technology. Even then, the experimental results are based on 4x4 crossbars.

Unlike what the authors claim, this is still at the level of laboratory experiments, with some basic demonstration. PUFs with memristors of course is not new, as the authors are aware, similar approaches, for example in Ref. 26 have implemented security primitives with PUFs.

Here, the authors make some claims about exposed keys and reconfigurability, but even if these are accurate, it is unclear why a 4x4 PUF with some reconfigurability is such a major advance or why it points to maturity. In many cases with PUFs, it is unclear to me why one would need to refresh the key so much, or why these prior approaches

could not refresh their keys (not explained well), or why this is so important to advance the field.

Answer: First, we want to thank the reviewer for such an extensive review and his/her expertise in the field. We believe Reviewer 1's skepticism represents most of the readers. Therefore, we carefully incorporated these feedback and criticisms.

First, we removed the claim of "maturity" of memristor technology since it seems to have misled the reviewer. The term seems to have implied full technological readiness, which we did not intend. Instead, we wanted to emphasize functional flexibility, and we hope it will be apparent to the reviewer from our following responses and the revised manuscript.

The reviewer raised three main issues: PUF generation is not new, Boolean logic is not new, and the performance evaluation is questionable.

We agree that PUF generation by itself is not a functionally new primitive. However, our PUF generation scheme is advantageous over previous works and pushes the boundaries of state-of-the-art memristor research. To illustrate the advantages better, we included Table 1 in the main text (reproduced below). Our system offers reconfigurability, concealability, nonvolatility, passive nonvolatile crossbar array implementation, and logic operation capability. Each item has been demonstrated individually using memristor-based approaches, but due to various challenges (outlined in the text), no single approach has achieved all these features simultaneously. As to why certain features may be helpful, such as reconfigurability (as the reviewer questioned), we elaborated on it in the main text.

To address the reviewer's criticism that a 4×4 array is not a significant demonstration, which we agree, we performed additional circuit simulations to study the scalability of our arrays (Supplementary Note 2). Specifically, we studied the effects of the sneak paths and line resistance, which indicated scalability of up to 400×400 (based on the worst-case scenario of the sneak paths) without significant degradation in storage capacity. Furthermore, while a 4×4 demonstration is indeed relatively small, several introductions of a qualitatively novel scheme have been done on small arrays (Lee, S. et al. *Adv. Mater.* 34, 2203558 (2022), Zhang, R. et al. *Nanoscale* 10, 2721 (2018)). We believe that demonstrating the new features on a 4×4 array verified our approaches.

Specific changes:

- Removed the claim of "maturity" in abstract (page 1) and introduction (page 2).

(Modified, abstract) Using these computing primitives, this work experimentally demonstrates a single system that performs cryptographic key generation, universal Boolean logic operations, and encryption/decryption. Circuit-based calculations reveal the energy and latency advantages of the CuTeHO memristors in these operations. This work illustrates the functional flexibility of memristors in implementing operations with varying component-level requirements.

(Added, main text) Thus, despite the high energy efficiency of memristor-based computing systems demonstrated for specific applications, they still fail to compete with highly flexible digital processors.

This work demonstrates a highly functional scheme of memristor-based hardware that integrates security, computing and memory capabilities using the $\text{Cu}_{0.3}\text{Te}_{0.7}/\text{HfO}_2$ (CuTeHO) ion-migration-driven memristor. Its stochastic switching variation is exploited as an entropy source for generating PUFs, with the added features of concealability and reconfigurability. When operated away from biases that promote stochasticity, a single CuTeHO memristor device can be guided via a sequence of up to three switching operations to exhibit all Boolean logic operations. Furthermore, this work demonstrates multiple computing processes performed on memristive hardware by exploiting the dual functional nature of the CuTeHO memristors: unclonable key generation, logic operations, and encryption/decryption, thereby demonstrating functional flexibility that has been missing in memristor-based systems. The energy consumption and latency of these operations are notably lower compared to prevailing digital computing approaches. While there are also performance advantages (compared to prior works that used memristors) in both PUF generation and Boolean logic, the novelty of this work is the functional ability to switch between them on demand. Such flexibility enables an essential step towards functionally dense post-digital computers. As a demonstration of the utility and effectiveness of the encryption-decryption scheme, using crossbar arrays constructed using different nonvolatile memristors, matrix multiplication operations are performed on encrypted data without a noticeable loss of information.

- Reference to scalability studies on page 6.

(Added, main text) To further validate the practical use of the device in a large-scale array, using circuit-based simulations, sneak currents and line resistance issues were evaluated as the array size was increased (Supplementary Note 2). With the sneak current suppressed by the rectifying behavior, no critical issues were observed in array sizes up to 400×400 in the worst-case scenario and 1000×1000 when half the devices were at HRS (Supplementary Fig. 4). The line resistance did not affect the operation in a large-scale array (Supplementary Fig. 5).

- Supplementary Note 2: details of the scalability studies (sneak paths and line resistance).
 - Supplementary Figs. 3-4 (reproduced below).

Supplementary Note 2. Scalability of the crossbar array.

2.1. Sneak current issue depending on the size of the array.

A CuTeHO single device was modeled, as shown in Supplementary Fig. 4a, to evaluate the possible negative impact of sneak current on the accuracy of current reading when crossbar array size increased. Supplementary Fig. 4b quantitatively demonstrates the rectifying ratio of the CuTeHO device. For the worst-case scenario to maximize the influence of sneak current, only the (N, N) component was settled to HRS, while the remaining

components were settled to LRS (N represents the number of wires in one dimension, Supplementary Fig. 4c). Supplementary Fig. 4d shows the variation in the current passed through the HRS device at (N, N) position as N increased from 2 to 1000, under the given circumstance. According to the simulation results, there was no significant change in the HRS output current up to N of 400 due to the excellent rectifying characteristic of the device. The inset of Supplementary Fig. 4d shows that the sneak current necessarily includes a path from the bottom electrode to the top electrode of the device, which the rectifying property of the CuTeHO device could suppress. However, when N exceeded 400, the worst-case HRS output current exceeded the LRS output current, leading to misinterpretation of the device's state. However, the N could be increased to 1000 when half the devices were at HRS.

2.2. Impact of line resistance on the crossbar array.

The influence of line resistance was also analyzed. Supplementary Fig. 5a represents the crossbar array with the line resistance when N is 100. To maximize the impact of line resistance (green area), only the CuTeHO component (red area), the most influenced cell by the line resistance, was set as LRS and read. Supplementary Fig. 5b shows that the total resistance is affected only when the line resistance becomes unrealistically large because even the LRS resistance of the CuTeHO device is much higher than the line resistance.

Supplementary Fig. 4 | **a**, I-V curves of the device with the fitting results (red dashed line). **b**, On/off and rectifying ratios with varying read voltages. **c**, Schematic diagram of the crossbar array. Red colors are LRS devices. **d**, The current passing through the HRS device at (N, N) position.

Supplementary Fig. 5 | a, Schematic diagram of the crossbar array at $N = 100$. b, R_{total} at different line resistance values.

- Pages 6 and 10: additional notes on the need for reconfigurability and the utility of rectifying behavior in implementation in passive crossbar networks.

(Added, main text) In contrast, the intrinsic stochasticity of memristors during their switching process holds great promise for hardware security applications owing to their high switching speed, low power consumption, and remarkable potential for scalability^{25–30}. A critical requirement for modern PUFs is reconfigurability, where keys (the challenge-response pairs) are regenerated within the same chip to account for the increasing probability of information leakage with time. A second critical requirement is the concealability of the key when stored in hardware. There have been reports on reconfigurable memristive PUFs^{26–29}, for instance, using conductance variations of neighboring cells to achieve reconfigurability. However, these reconfigurable memristive PUFs were not area-efficient as they required at least two memristors to generate a single bit. Furthermore, there has been a report of the concealment of keys in a memristive PUF, where the data was concealed by switching all devices into the LRS³⁰. However, no memristive PUF system has offered both concealability and reconfigurability.

(Added, main text) Finally, Table 1 shows the comparison with other memristive PUFs demonstrating the advancements in this work. This work provides reconfigurability, concealability, and area efficiency (a single cell represents a full bit). The device's self-rectifying behavior also enables its implementation in selectorless (passive) crossbar array configuration, whereas prior works required complex design using transistors or selectors. As discussed in the next section, the CuTeHO device has the unique ability to perform Boolean logic operations in addition to PUF generation.

- Table 1: Performance comparison of memristor-based PUFs.

Table 1 | Comparison of memristor-based PUFs.

	This work	Zhang, et al. ³¹	Nili, et al. ³²	Jiang, et al. ³³	John, et al. ²⁸	Gao, et al. ³⁴
Device structure	Simple (Cu _x Te _{1-x} /HfO ₂ /Pt)	Simple (Au/Ag:SiO ₂ /Au)	Complex (Pt/TiN/TiO _{2-x} /Al ₂ O ₃ /Pt/TiN/TiO _{2-x} /Al ₂ O ₃ /Pt/TiO _{2-x} /Al ₂ O ₃ /Pt/TiO ₂ /Al ₂ O ₃)	Simple (Ta/HfO ₂ /Pt)	Complex (Ag/PMMA/PrPyr[PbI ₃]/PEDOT:PSS/ITO)	Simple (TiN/TaO _x /HfO _x /TiN)
Bits per cell	1	1	1 / # of rows	0.5	0.5	1 (needs extra current source)
PUF methodology	SET voltage variations	Device switchability based on Ag concentrations	I-V nonlinearity variations	LRS of neighboring cells	HRS of neighboring cells	HRS compared to a reference
Uniqueness (%)	50.62	50.68	49.96	50.06	49.02	50.52
NIST test	Passed	Not reported	Passed	Not reported	Passed	Passed
Reconfigurability	Yes	No	Yes	Yes	Yes	No
Concealability	Yes	No	No	No	No	Yes
Nonvolatility	Yes	No	Yes	Yes	Yes	Yes
Logic operation capability	Yes	No	No	No	No	No
Implementable with passive nonvolatile crossbar	Yes	No	No	No	No	No

Next, the authors move on to some basic logic gates with new devices. Once again, much of their experimental results are extremely basic (and hence I find their GPU/CPU comparisons disingenuous). These functions are performed with hundreds of billions of transistors in CPUs and GPUs whereas in FIG. 3 we are observing some basic finite state machine (FSM) with a handful of states (less than 10) and somehow we are to believe in the maturity of the sequential logic capabilities of the memristive approach.

The device community has largely moved on from trying to implement basic Boolean logic with new devices because CMOS transistors are impossible to beat in this space. The authors can check Dmitri Nikonov's work on benchmarking Boolean logic with a variety of new technologies. Unfortunately, the implementation of a 4x4 PUF or a 4-state FSM do not seem like the "end-to-end" solution this paper claims.

Answer: We appreciate this critical comment again. We reviewed Dmitri Nikonov's recent works (e.g., Nikonov et al. *IEEE J. Exploratory Solid-State Computational Devices and Circuits*, "Benchmarking of Beyond-CMOS Exploratory Devices for Logic Integrated Circuits", 1, 3 (2015), Nikonov, D. E. et al. *IEEE. Nanotechnology Magazine*, "Review of Simulation Methods for Design of Spin Logic", 17, 4 (2023)). Those articles highlighted the urgent necessity of beyond-CMOS devices in building logic-integrated circuits. They also presented a benchmarking framework based on simulated works, noting that most new device concepts are based on simulations, which might be indispensable for justifying the feasibility of new concepts.

In our work, we provided experimental results to substantiate the viability of post-CMOS devices in building universal logic gates. Still, we agree with the reviewer that our demonstration is only basic, and actual technological readiness will require demonstrations of extreme scales, manufacturability and reliability. However, we again want to emphasize that our demonstration establishes the functional flexibility of our memristors in performing not only PUF generation but also logic operations, and this functional flexibility has not been demonstrated before. This demonstration of dual functions is exciting since the two functions have opposing requirements (PUF requiring unpredictability and logic requiring predictability).

These issues are now emphasized in the revised text. Also, we showed that our demonstration of logic operations is better than any prior demonstration using memristor technology (in the form of a full adder) (Supplementary Tables 2-3). While this is not with full technological maturity, it is still an advancement over the state-of-the-art memristor research.

As stated in our response above, we now call it functional flexibility instead of calling this technological maturity.

Specific changes:

- Emphasis on efficient logic operations on Page 13.

(Added, main text) All 16 logic operations are experimentally demonstrated by sequentially applying voltage conditions specific to each logic operation in no more than three steps to substantiate the viability of the suggested concept further (Fig. 3e and Supplementary Figs. 14-19).

- Emphasis on adder/subtractor constructed using logic based on our memristors being efficient compared to other memristor-based approaches (Page 13).

(Added, main text) Using the suggested DC logic method, a 1-bit binary full adder/subtractor was constructed using a 2×2 crossbar in 8 steps (Supplementary Notes 3 and 4, Supplementary Figs. 21-23). Utilizing the

structural and functional characteristics of the crossbar, this work shows that the number of required devices and operation steps are lower than previously reported memristor-based adders (Supplementary Table 2). Especially, the suggested method has a significantly lower energy cost than alternatives. Furthermore, the adder and subtractor can be combined in a single 3×2 memristive crossbar, allowing concurrent mapping of outputs for both operations (Supplementary Note 4.3, Supplementary Figs. 24-27 and Supplementary Table 3).

What follows is a basic encryption/decryption demo with slightly larger arrays of memristors preceding a Memristor / CPU / GPU table comparison.

I found it hard to follow how this table is obtained, since the numbers are well-hidden in a disorganized 62-page supplementary.

Take RNG of CPU for example. The paper the authors cite from Intel can generate random numbers with a throughput of 323 Gbit per second. In any real application that needs RNGs, requiring trillions and trillions of random numbers are very common. What is the throughput of the memristor approach that leads to 51.7 pJ of energy consumption? How are we sure that the size, throughput and statistical properties of RNGs are equivalent? I see some NIST test comparisons, but I am curious whether the authors really believe that their approach on all of these "training, RNG, Inference, Encryption" functions can really outperform CPUs and GPUs, because, as they would appreciate, winning against CPUs or GPUs in ONE of these applications convincingly would be a win for the field.

Answer: We understand the reviewer's main criticism that our performance projections are too optimistic. That performance table was made using projections from multiple memristor technologies (e.g., our projections of the CuTeHO memristors for RNG and encryption/decryption, while we used other published estimates using other memristor technologies for inference, training, etc.). We acknowledge that such an optimistic projection can mislead the reader into believing that our approach alone can achieve superlative performance on every function listed. In the revised main text, we limited to projecting the performance of only the experimentally demonstrated functions: RNG and encryption/decryption (Table 2). We provided reasonably detailed calculations in the supplement (Supplementary Figs. 34 and 35, Supplementary Note 8).

The Intel paper referenced by the reviewer produces 323 Gbps/W (not only a bit rate but a bit rate per watt), which is 3 pJ/bit. We elaborated on a comparison to this paper in Supplementary Note 8.1. In short, our calculated/projected bit rate is close to 500 Gbps/W (2 pJ/bit). On the statistical quality of the random numbers, we performed all the NIST randomness tests, which our experimental results passed. Although other tests exist, this method is regarded as standard for RNGs. We used published memristor performance estimates for scaled devices (down to 16 nm). In the revision, we also clearly acknowledged the possible sources of deviations from the performance comparisons and that the quantitative results must be taken with a grain of salt. Notwithstanding the comment, we firmly believe the approach here is highly competitive in the post-CMOS world.

We apologize for the supplement being disorganized. We strictly followed the journal's format to keep the figures, tables and notes separated. This format exhibits scalability issues as the size of the supplement grows. We revised

the structure to a standard narrative (with relevant text and figures grouped). We hope it is improved now.

Specific changes:

- Simplified Table 2.

Table 2 | Energy and latency comparisons of XOR encryption (encrypting 'A'-shaped data with PUF-1, as shown in Supplementary Fig. 29) between the memristor-based approach and CMOS-based computers.

ⁱ: Supplementary Note 8.1 and Figure S34, ⁱⁱ: Figure S34 for memristor, ⁱⁱⁱ: Supplementary Note 8.2 and Figure S35 for CMOS computer.

	Memristor		CMOS computer	
	Energy	Latency	Energy	Latency
RNGⁱ	34.4 pJ	23.5 ns	0.14 nJ	0.1 μs
Encryption/decryption^{ii,iii}	0.1 pJ	54.4 ns	2.88 nJ	0.14 ns
SUM	0.13 nJ	77.9 ns	3.02 nJ	0.1 μs

- Additional explanation in Supplementary Note 8.

Supplementary Note 8. Energy and latency comparisons of memristor-based encrypted inference with traditional computing systems.

8.1. Energy/latency for key generation.

The performance of CMOS-based digital computers for random number generation (RNG) is estimated from a recent work by Intel²³. They demonstrated various operation modes, each being optimized for a different quantity. The standard mode produced 162.5 Mb/s, with 1.5 mW of power consumption, yielding an energy efficiency of 9 pJ/bit. Mode (vi) was optimized for energy efficiency, with a peak encrypted bit throughput of 323 Gbps/W (or 323 Gb/J), with an energy efficiency of 3 pJ/bit. Thus, in mode (i), 144 pJ is consumed during 94 ns to generate 16 random binary numbers.

The energy and latency calculations (Supplementary Fig. 34) of PUF-1 for the memristor were performed based on our hardware setup described in Supplementary Fig. 30.

8.2. Energy/latency for encryption/decryption by XOR.

Nguyen et al. performed computer architectural studies for the energy benchmarking of CMOS-based XOR gate operation^{24,25}. The 4-bit carry-lookahead adder (CLA) in Supplementary Fig. 35a constitutes the basic building block of the simulated CMOS-based 32-bit parallel adders with 8 XOR gates, 14 AND gates, and 4 OR gates (including a two-level carry-lookahead box with dotted line). Eight rippled-4-bit CLAs form a 32-bit adder (Supplementary Fig. 35b), of which 32 replications and 8 kB L1 cache memory (Supplementary Fig. 35c), form a cluster to perform 32 additions simultaneously. One cluster with 32 cores consumes 1.2 μ J from the computer architecture simulation²⁵, and this value was adopted to calculate the energy consumption of a logic gate from CMOS transistors assuming similar energy consumptions (6 transistors per gate regardless of XOR, AND, and OR). Therefore, the number of logic gates per cluster is 6656 (= 32 adders \times 8 CLAs \times 26 gates), and the energy consumption per logic gate is 0.18 nJ. The number of total XOR operations used in the present demonstration ('A'-shaped data) is 16, so the total energy consumption from the encryption/decryption is 2.88 nJ. The latency for one XOR operation is 9 ps (0.14 ns for 16 XOR operations). **For XOR encryption/decryption using the memristive system, the energy and latency calculations of encrypting 'A'-shaped data were performed based on the hardware setup described in Supplementary Fig. 30 (Supplementary Fig. 34). Table 2 summarizes the performance associated with the XOR encryption.**

Overall, I find the paper highly overstating what it actually delivers. Unfortunately the comparisons seem unconvincing and superficial. A much more tempered manuscript could deserve publication, but I am not sure if in Nature Communications.

Answer: We toned down some of the claims (such as the approach demonstrating technological maturity) and removed some quantitative calculations (e.g., on inference and training). Still, we firmly maintained the clear benefits and advancement of the approach presented here, using experimental demonstrations and reasonable performance projections. We hope the reviewer appreciates these revisions.

Reviewer #2:

Author(s) presented good work in the domain of encryption and computing simultaneously with CuTeHO based stochastic memristors. The whole manuscript is well organized and useful results are demonstrated. This work can be considered but few points need to be addressed.

1. Clearly explain the variability of molar fraction 'x' in the development of PUF and computing system.

Answer: Thank you for the careful comment. We fixed the intermediate stoichiometry of $x = 0.3$ for all the applications. We added the following comments to make this point clear.

Specific changes: Clarification of $x = 0.3$ in the rest of the paper (Page 6).

(Added, main text) The volatile and nonvolatile behaviors for the cases of $x = 0.1$ and $x = 0.6$ are not surprising as reported elsewhere²⁰⁻²³. However, the dual behavior at an intermediate stoichiometry ($x = 0.3$) is unexplored and is utilized for diverse applications in this work, which are explained in the following sections.

2. Give more clarity of deep learning model used for encryption of data.

Answer: Thank you for the feedback. The reviewer might have missed a detailed description of our deep neural network in Supplementary Note 6 due to the absence of a note in the main text. Hence, we added an additional description of the network in the main text as well as directing readers to Supplementary Note 6 (in Page 17)

Specific changes: Detailed description of deep neural network (Page 17).

(Added, main text) Then, A^* was fed into a trained neural network with weights W , resulting in a dot product output $A^* \cdot W$. A two-layer network (24 input, 33 hidden, and 7 output neurons) was employed, which was trained to classify seven distorted braille words (Figs. 4g and h). Identical weights trained in the previous work in the Hopfield network structure⁴⁵ were deployed for this encrypted pattern classification. Detailed information is available in Supplementary Note 6. The output of the neural network was then decrypted into $A \cdot W$ via $A \cdot W = A^* \cdot W - k \cdot W$.

Reviewer #3:

This paper's biggest claim is that they have created a novel memristor that they can use both for computing and security purposes. The authors claim that under specific biases, they can either induce the stochastic behavior required to have a memristor be able to generate truly random numbers (TRNG) and unclonable functions (PUFs) or perform universal Boolean logic. The trick is to tune the strength of the conductive filament, which stays weak (volatile) when the current is under 10 nA and nonvolatile if the current is above 100 nA.

The device consists of a 50 nm Cu/Te top electrode, a 10 nm HfO for their switching layer, and a 50 nm Pt for the bottom electrode. The authors settled on a 30% concentration of Cu to Te for the top electrode as this device has nonvolatile characteristics and self-rectifying behavior. The operating range for the device seems to be -3.5 V for reset and 2.5 V for set (they have shown extensive statistical data to arrive at those values). A weak conductive filament formed when the compliance current I_{cc} was set to 10 nA, which could be easily broken with a slight reset. Those fractions from the broken conductive filament serve as traps to induce the stochastic behavior necessary for PUFs.

When the device is operated at 100 nA I_{cc} , this stochastic behavior doesn't exist as the conduction filament is much stronger and more challenging to break. The authors experimentally demonstrated dual conditional logic operations and encryption and decryption.

The authors had extensive data to support most of the claims. The supplemental file is filled with stability, cycle-to-cycle, pulse behavior, and retention data on the device. Their claims were all backed up with experimental data, not simulated.

However, there are some critical issues that should be addressed. First, the device physics component is relatively weak, with only a handwaving explanation. Some level of physical characterization and simulation are necessary to strengthen this part.

Answer: Thank you for the detailed comment. Our electrical characterization shows evidence about the device's switching mechanism, but the information was not detailed enough. In order to verify its complex switching mechanism, we controlled two variables: Cu composition and compliance current.

Varying the Cu composition in the $\text{Cu}_x\text{Te}_{1-x}$ electrode resulted in different behaviors, ranging from volatile (at $x = 0.1$) to nonvolatile (at $x = 0.6$), as demonstrated in Figs. 1a-c. It was previously reported that the filament's stability is strongly influenced by its surface curvature (Woo, K. S. et al. *Adv. Intell. Syst.* 3, 2100062 (2021); Wang, W. et al. *Nat. Commun.* 10, 81 (2019); Hsiung, C.-P. et al. *ACS Nano* 4, 5414 (2010)), which determines the device to be either a diffusive memristor (with filament rupture at zero bias) or a CBRAM (where the filament is maintained at zero bias). This model explains two disparate behaviors at $x = 0.1$ and 0.6. However, at $x = 0.3$, the device showed nonvolatile and rectifying behavior, which is not the case for CBRAM. Self-rectifying behavior is the common characteristic observed in the electron trapping/detrapping mechanism, indicating a different switching mechanism following slight filament rupture. We would expect CBRAM behavior with substantial current flow in the negative voltage region if the switching mechanism was purely based on Cu conductive filaments. To further investigate the switching behavior at $x = 0.3$, we varied the compliance current to observe the switching behavior with different filament sizes. Similarly, we observed volatile behavior at $I_{cc} = 50$ pA (Supplementary Fig. 12b) and CBRAM behavior at $I_{cc} = 100$ nA (Supplementary Fig. 2b). Given the small internal electric field due to the

work function difference between $\text{Cu}_{0.3}\text{Te}_{0.7}$ electrode and the remaining Cu filament, electron trapping/detrapping seems plausible.

As the reviewer requested additional data, we also measured the I - V curves of the LRS device at different ambient temperatures. There was no current decrease in the LRS device with increasing temperature, indicating that the conduction is not based on a purely metallic filament. We added texts and figures for a more detailed explanation of the device's switching mechanism.

Specific changes

- Additional explanation in Page 5 on the switching mechanism.

(Added, main text) In contrast, the $\text{Cu}_x\text{Te}_{1-x}\text{HO}$ memristor with $x = 0.6$ ($\text{Cu}_{0.6}\text{Te}_{0.4}/\text{HfO}_2$), which forms even stronger Cu conductive filaments (CFs), exhibits conductive bridge random access memory (CBRAM) behavior with Ohmic conduction in a low resistance state (LRS) (Fig. 1c and Supplementary Fig. 2a). Previous studies reported that the filament's stability is strongly influenced by its surface curvature¹⁰⁻¹⁴, which determines the device to be either a diffusive memristor or a CBRAM. However, at $x = 0.3$, the device exhibits nonvolatile and rectifying behavior, which deviates from the typical characteristics of CBRAM. It exhibits an Ohmic conduction property immediately after being switched to the LRS by the I - V sweep (Curve 1 in Fig. 1b). However, the subsequent sweep does not exhibit Ohmic conduction, suggesting the potential presence of a distinct switching mechanism (Curve 2). The conduction behavior observed in the LRS and the rectification characteristics in the negative bias region (Curve 3) are similar to that of an electronic switching memristor, where electron trapping and detrapping occur at a trap layer¹⁵⁻¹⁸.

Additionally, the I - V characteristics of the CuTeHO memristor were examined at a higher I_{cc} of 100 nA (Supplementary Fig. 2b), where sufficiently strong CFs form, which cannot be ruptured at zero bias. Similar to the $\text{Cu}_{0.6}\text{Te}_{0.4}/\text{HfO}_2$ device, no rectifying behavior was found due to the stronger CF at higher I_{cc} . Furthermore, the I - V curves of the LRS device were measured at different temperatures (Supplementary Fig. 3), where the absence of a decrease in the current in the LRS device with increasing temperature indicates that the conduction is not solely based on a metallic CF.

Based on the varying qualitative memristive behaviors, the switching mechanism of the CuTeHO memristor can be understood as a combination of CF formation/rupture and electron trapping/detrapping, as illustrated in Fig. 1d. The relatively weak Cu CFs are formed during the SET (switching from high resistance state (HRS) to LRS) process. At zero bias, the diffusion of the Cu atoms from the CF to the surrounding HfO_2 region causes a slight rupture of the filament (similar to the behavior observed in the $\text{Cu}_{0.1}\text{Te}_{0.9}/\text{HfO}_2$ device), simultaneously creating a localized path for charge trapping. In other words, the residual Cu CFs near the bottom electrode (BE) can be considered an extended electrode. The electron trapping/detrapping process in the local CF-ruptured region mediated by traps at roughly 0.7 eV below the conduction band (Fig. 1e), reported elsewhere¹⁸, can explain the observed switching behaviors. Kim *et al.*¹⁹ confirmed this switching mechanism, where the insulator was divided into switching and filament regions. The self-rectifying behavior in the negative bias region is attributed to the slightly ruptured filament and a possible barrier formed at $\text{Cu}_x\text{Te}_{1-x}/\text{HfO}_2$ interface due to the Fermi level pinning.

- New Supplementary Figure 3 (reproduced below).

Supplementary Fig. 3 | a, I-V curves of the LRS device with varying temperatures. b, Resistance of the LRS device at 0.8 V.

Second, the sneak path issue will be a showstopper for scaling up the CuTeHO devices technology. It is suggested that the authors provide an analysis of how large an array can be operational for logical operations.

Answer: We agree that the sneak path issue can severely limit the array sizes and thus limit the technological viability of this approach. To evaluate this issue, we performed circuit simulations using parametrized models of the devices, along with realistic line resistances, for varying array sizes. We found that prohibitive limit (e.g., such as the sneak currents being as large as the low-resistance currents), even in the worst case, was reached only at an array size $> 400 \times 400$, while for the case of half the devices at HRS, the limit was $> 1000 \times 1000$, which is compatible with state of the art. We also evaluated the effect of line resistance on the array's storage capacity. The line resistance did not affect the storage capacity unless it was assumed to be unusually large, mainly owing to the inherently large LRS's resistance levels.

Specific changes

- Reference to scalability studies in page 6.

(Added, main text) To further validate the practical use of the device in a large-scale array, using circuit-based simulations, sneak currents and line resistance issues were evaluated as the array size was increased (Supplementary Note 2). With the sneak current suppressed by the rectifying behavior, no critical issues were observed in array sizes up to 400×400 in the worst-case scenario and 1000×1000 when half the devices were at HRS (Supplementary Fig. 4). The line resistance did not affect the operation in a large-scale array (Supplementary Fig. 5).

- Supplementary Note 2: details of the scalability studies (sneak paths and line resistance).
 - Supplementary Figs. 3-4 (reproduced below).

Supplementary Note 2. Scalability of the crossbar array.

2.1. Sneak current issue depending on the size of the array.

A CuTeHO single device was modeled, as shown in Supplementary Fig. 4a, to evaluate the possible negative impact of sneak current on the accuracy of current reading when crossbar array size increased. Supplementary Fig. 4b quantitatively demonstrates the rectifying ratio of the CuTeHO device. For the worst-case scenario to maximize the influence of sneak current, only the (N, N) component was settled to HRS, while the remaining components were settled to LRS (N represents the number of wires in one dimension, Supplementary Fig. 4c). Supplementary Fig. 4d shows the variation in the current passed through the HRS device at (N, N) position as N increased from 2 to 1000, under the given circumstance. According to the simulation results, there was no significant change in the HRS output current up to N of 400 due to the excellent rectifying characteristic of the device. The inset of Supplementary Fig. 4d shows that the sneak current necessarily includes a path from the bottom electrode to the top electrode of the device, which the rectifying property of the CuTeHO device could suppress. However, when N exceeded 400, the worst-case HRS output current exceeded the LRS output current, leading to misinterpretation of the device's state. However, the N could be increased to 1000 when half the devices were at HRS.

2.2. Impact of line resistance on the crossbar array.

The influence of line resistance was also analyzed. Supplementary Fig. 5a represents the crossbar array with the line resistance when N is 100. To maximize the impact of line resistance (green area), only the CuTeHO component (red area), the most influenced cell by the line resistance, was set as LRS and read. Supplementary Fig. 5b shows that the total resistance is affected only when the line resistance becomes unrealistically large because even the LRS resistance of the CuTeHO device is much higher than the line resistance.

Supplementary Fig. 4 | **a**, I-V curves of the device with the fitting results (red dashed line). **b**, On/off and rectifying ratios with varying read voltages. **c**, Schematic diagram of the crossbar array. Red colors are LRS devices. **d**, The current passing through the HRS device at (N, N) position.

Supplementary Fig. 5 | **a**, Schematic diagram of the crossbar array at $N = 100$. **b**, R_{total} at different line resistance values.

Third, the benchmarking is not specific to certain models of GPU and CPU, and it is vague if the external PCB boards are included in the benchmarking.

Answer: We simplified Table 2 (performance benchmarking) in two ways. First, we focused only on the functions that we experimentally demonstrated using the CuTeHO memristors. We removed the performance estimates for inference and training, which were performed using other memristors. We hope this revision aligns with the reviewer's suggestion (in the next comment) on not including encrypted inference on the 1T1R array. Second,

instead of two separate entries for CPU and GPU, we used a single CMOS-based system as a benchmark, using published performance estimates based on specific architectures optimized for either of the two functions (XOR and RNG). In supplementary Note 8, we have provided additional details.

Specific changes:

- Simplified Table 2.

Table 2 | Energy and latency comparisons of XOR encryption (encrypting 'A'-shaped data with PUF-1, as shown in Supplementary Fig. 29) between the memristor-based approach and CMOS-based computers.

ⁱ: Supplementary Note 8.1 and Figure S34, ⁱⁱ: Figure S34 for memristor, ⁱⁱⁱ: Supplementary Note 8.2 and Figure S35 for CMOS computer.

	Memristor		CMOS computer	
	Energy	Latency	Energy	Latency
RNGⁱ	34.4 pJ	23.5 ns	0.14 nJ	0.1 μs
Encryption/decryption^{ii,iii}	0.1 pJ	54.4 ns	2.88 nJ	0.14 ns
SUM	0.13 nJ	77.9 ns	3.02 nJ	0.1 μs

- Additional explanation in Supplementary Note 8.

Supplementary Note 8. Energy and latency comparisons of memristor-based encrypted inference with traditional computing systems.

8.1. Energy/latency for key generation.

The performance of CMOS-based digital computers for random number generation (RNG) is estimated from a recent work by Intel²³. They demonstrated various operation modes, each being optimized for a different quantity. The standard mode produced 162.5 Mb/s, with 1.5 mW of power consumption, yielding an energy efficiency of 9 pJ/bit. Mode (vi) was optimized for energy efficiency, with a peak encrypted bit throughput of 323 Gbps/W (or 323 Gb/J), with an energy efficiency of 3 pJ/bit. Thus, in mode (i), 144 pJ is consumed during 94 ns to generate 16 random binary numbers.

The energy and latency calculations (Supplementary Fig. 34) of PUF-1 for the memristor were performed based on our hardware setup described in Supplementary Fig. 30.

8.2. Energy/latency for encryption/decryption by XOR.

Nguyen et al. performed computer architectural studies for the energy benchmarking of CMOS-based XOR gate operation^{34,35}. The 4-bit carry-lookahead adder (CLA) in Supplementary Fig. 35a constitutes the basic building block of the simulated CMOS-based 32-bit parallel adders with 8 XOR gates, 14 AND gates, and 4 OR gates (including a two-level carry-lookahead box with dotted line). Eight rippled-4-bit CLAs form a 32-bit adder (Supplementary Fig. 35b), of which 32 replications and 8 kB L1 cache memory (Supplementary Fig. 35c), form a cluster to perform 32 additions simultaneously. One cluster with 32 cores consumes 1.2 μ J from the computer architecture simulation³⁵, and this value was adopted to calculate the energy consumption of a logic gate from CMOS transistors assuming similar energy consumptions (6 transistors per gate regardless of XOR, AND, and OR). Therefore, the number of logic gates per cluster is 6656 (= 32 adders \times 8 CLAs \times 26 gates), and the energy consumption per logic gate is 0.18 nJ. The number of total XOR operations used in the present demonstration ('A'-shaped data) is 16, so the total energy consumption from the encryption/decryption is 2.88 nJ. The latency for one XOR operation is 9 ps (0.14 ns for 16 XOR operations). **For XOR encryption/decryption using the memristive system, the energy and latency calculations of encrypting 'A'-shaped data were performed based on the hardware setup described in Supplementary Fig. 30 (Supplementary Fig. 34). Table 2 summarizes the performance associated with the XOR encryption.**

Finally, the 1T1R array part is disjointed from the main idea of the manuscript. It is not the CuTeHO device, but an existing cell technology from Applied Materials. Similar type inference engines (with much larger scale) have been demonstrated in both academia and industry. It is an unnecessary decoration, and I suggest the authors remove this part of irrelevant data.

Answer: We agree that the demonstration of inference (matrix multiplication) on encrypted data on the 1T1R array is unrelated to the CuTeHO devices presented here and uses a reasonably well-established crossbar array. We included it to show that the encrypted data provided by the CuTeHO devices can undergo additional operations in standard AI systems without losing the original data. It may appeal to more readers to see a full AI-style application compatible with a new encryption scheme. We are open to removing this part but seek the reviewer's and editor's opinions.

REVIEWER COMMENTS

Reviewer #1 (Remarks to the Author):

I greatly appreciate the revisions the authors have performed. As I carefully explain below however, in my view, the paper comes close to the very high standard the community assigns to Nature Communications and therefore I cannot recommend publication.

In the previous round, I had three main criticisms which the authors took to heart:

(1) The lack of novelty in PUF generation.

(2) The difficulty of conventional Boolean logic with novel nanodevices when measured against the impossibly high-standards of modern CMOS technology

(3) Performance evaluations in certain applications such as Random Number Generation.

The authors, to their credit, took these criticisms seriously and did what they could to respond to them. However, I find that in each case the authors agreed to the severity of the criticism, and in each case, I feel that the manuscript fell short of what the broad community of Nature Communications would expect.

Crucially, a paper of this caliber should extend the state-of-the-art in nanoelectronics and not in the limited scope of memristor research (please see below).

For (1), they now show a clear Table illustrating how their approach differs from the state of the art. Examining the table, I see that the differences with prior art are not that much, and the authors take credit for "logic operation capability" but as I argued before (and now), I do not think this capability offers any real (remotely) competitive advantages against Boolean logic with CMOS. As such, the lack of novelty remains in my view.

For (2), the authors agree that compared to the state-of-the-art Boolean logic, their demonstration remains primitive. I am all for primitive device concepts if they offer future promise, but in this case there is not even a Nikonov style benchmarking effort that would show how this primitive technology would scale.

The authors argue that while this is not competitive against CMOS (in theory or in practice), they are competitive against all other memristors approaches.

I am sorry but this is a very poor way of arguing why this work which brags "flexibility" of Boolean logic should be of broad interest to the community. Yes, these results may be better than all prior memristor proposals for Boolean logic, but that still doesn't change the fact that Boolean logic with memristors lags

far behind the state-of-the-art and may sadly never materialize.

I cannot find such a line of reasoning convincing.

(3) Finally, the authors respond to my criticism of their random number generation with some projections. Here, too, unfortunately, one must compare an actual chip from Intel, to the author's projections. I have no doubt these projections were prepared earnestly and with reasonable assumptions, it seems unfair to compare them to an actual demonstration.

Once again: I am not against primitive demonstrations that exhibit important, potentially ground-breaking phenomena if scaled. In this case, however, I do not see the novelty or the promising avenues this research opens up for it to be featured on the pages of Nature Communications.

Reviewer #2 (Remarks to the Author):

queries are addressed properly

Reviewer #3 (Remarks to the Author):

In the previous round of review, we made the following four main suggestions for the authors to consider.

1. Explain the device physics as their part was rather weak.
2. Address the sneak path issue with the size of the array.
3. Better address the benchmarking as the benchmark comparison table for energy consumption was extremely vague and didn't include the peripheral circuitry.
4. Remove the 1T1R array data as it was irrelevant.

Having read their response and seeing the changes implemented, we have our comments on each of the four points as follows.

1. They did a good job clearing the confusion, referencing other papers to further support their claim, and explaining the multi-switching mechanism that they are seeing. It is great that the authors added the last paragraph and the new supplementary figures showing the change in resistance value based on temperature, which supports that their device doesn't operate purely on the CBRAM (formation and rupture of metallic filaments), but a combination of filament formation/rupture and electron trapping/de-trapping. It is a bit misleading, though, that the authors claim it is a "distinct switching mechanism", as it is pretty common that a memristor has more than one active switching mechanism.
2. Regarding the sneak path, the authors conducted a simulation using parameters derived from the device on varying array sizes. They found that in the worst-case scenario (sneak path current = LRS current), they can support an array of 400 x 400. They also added figures showing those scalability and sneak path studies.

3. For the energy consumption benchmarking, the authors simplified Table 2, included a supplementary figure (30) to describe a fully encrypted memory-PUF implementation, and cited a random number generator paper from Intel. However, it does not clearly state if the peripheral circuits are included in the energy consumption estimation. A detailed energy consumption estimation with all components included should be presented.

4. We still think the 1T1R array data is irrelevant as it is from a different device. It will only mislead readers, thinking the array is from the CuTeHO devices.

Response to Referees

Black (italics): Reviewer comments, Blue: Author responses, Red: Modified text in the manuscript/supplement. Black (non-italic): original text in the manuscript/supplement.

Reviewer #2 (Remarks to the Author):

queries are addressed properly

Thank you for the positive feedback on our paper.

Reviewer #3 (Remarks to the Author):

In the previous round of review, we made the following four main suggestions for the authors to consider.

- 1. Explain the device physics as their part was rather weak.*
- 2. Address the sneak path issue with the size of the array.*
- 3. Better address the benchmarking as the benchmark comparison table for energy consumption was extremely vague and didn't include the peripheral circuitry.*
- 4. Remove the 1T1R array data as it was irrelevant.*

Having read their response and seeing the changes implemented, we have our comments on each of the four points as follows.

1. They did a good job clearing the confusion, referencing other papers to further support their claim, and explaining the multi-switching mechanism that they are seeing. It is great that the authors added the last paragraph and the new supplementary figures showing the change in resistance value based on temperature, which supports that their device doesn't operate purely on the CBRAM (formation and rupture of metallic filaments), but a combination of filament formation/rupture and electron trapping/de-trapping. It is a bit misleading, though, that the authors claim it is a "distinct switching mechanism", as it is pretty common that a memristor has more than one active switching mechanism.

Answer: We agree that a "distinct switching mechanism" may be misleading, so we modified the text as "the potential coexistence of another switching mechanism".

Specific changes: Changed the claim of "distinct switching mechanism" on page 5.

(Modified, main text) However, the subsequent sweep does not exhibit Ohmic conduction, suggesting the potential coexistence of another switching mechanism (Curve 2).

2. Regarding the sneak path, the authors conducted a simulation using parameters derived from the device on varying array sizes. They found that in the worst-case scenario (sneak path current = LRS current), they can support an array of 400 x 400. They also added figures showing those scalability and sneak path studies.

Answer: Thank you for acknowledging our additional work.

3. For the energy consumption benchmarking, the authors simplified Table 2, included a supplementary figure (30) to describe a fully encrypted memory-PUF implementation, and cited a random number generator paper from Intel. However, it does not clearly state if the peripheral circuits are included in the energy consumption estimation. A detailed energy consumption estimation with all components included should be presented.

Answer: Thank you for this comment. All the peripheral circuits were included in the calculations.

Specific changes: Clarification of the peripheral circuits included (caption of Table 2, and references to the specific Supplementary sections/figures) (page 17).

4. *We still think the ITIR array data is irrelevant as it is from a different device. It will only mislead readers, thinking the array is from the CuTeHO devices.*

Answer: We now agree with the referee. Figs. 4c-i and related texts are moved to the supplement.

Specific changes: We moved Figs. 4c-i and related texts to Supplementary Fig. 31 (page 51 of the supplement) and Supplementary Note 6 (pages 47~49 of the supplement).

Reviewer #1 (Remarks to the Author):

I greatly appreciate the revisions the authors have performed. As I carefully explain below however, in my view, the paper comes close to the very high standard the community assigns to Nature Communications and therefore I cannot recommend publication.

In the previous round, I had three main criticisms which the authors took to heart:

(1) The lack of novelty in PUF generation.

(2) The difficulty of conventional Boolean logic with novel nanodevices when measured against the impossibly high-standards of modern CMOS technology

(3) Performance evaluations in certain applications such as Random Number Generation.

The authors, to their credit, took these criticisms seriously and did what they could to respond to them. However, I find that in each case the authors agreed to the severity of the criticism, and in each case, I feel that the manuscript fell short of what the broad community of Nature Communications would expect.

Crucially, a paper of this caliber should extend the state-of-the-art in nanoelectronics and not in the limited scope of memristor research (please see below).

Answer: While we appreciate the reviewer's comments, we cannot agree with the skepticism about this work's novelty because many papers published in the same journal have shown improvements compared to other memristor technologies (and not necessarily overall CMOS technologies):

1. Kim, G. et al. *Nature Communications*, 12, 2906 (2021): Comparison of volatile-memristor-based TRNGs (*Supplementary Figure 3*).

2. John, R. A. et al. *Nature Communications*, 12, 3681 (2021): Benchmark comparison of memristor-based PUFs (*Supplementary Table 3*).

3. Jeon, K. et al. *Nature Communications*, 15, 129 (2024): Comparison of neural network applications using memristive crossbar arrays (*Table 1 of the main text*).

All these previous works have a somewhat limited scope than ours because they aimed at a single application, while this work shows various applications (PUF generation, logic operation, encryption/decryption). Therefore, we believe this work is of broad interest to the readers of *Nature Communications*.

Specific changes: We have cited these papers in our revision and added a Discussion section (page 17) to provide our perspective on why best-of-memristor technologies are indeed important to be published by the best scientific journals since they form the pipeline for manufacturable post-CMOS technologies. Further, we have discussed why functional reconfigurability is important.

For (1), they now show a clear Table illustrating how their approach differs from the state of the art. Examining the table, I see that the differences with prior art are not that much, and the authors take credit for "logic operation capability" but as I argued before (and now), I do not think this capability

offers any real (remotely) competitive advantages against Boolean logic with CMOS. As such, the lack of novelty remains in my view.

Answer: The PUF comparison table (Table 1) clearly shows a considerable difference compared to the prior works. All the prior works showed the inability to implement at least three items. In contrast, our work demonstrates all of them in a single system (reconfigurability, concealability, nonvolatility, implementation in a passive crossbar array and logic capability).

Gao et al. (Gao, B. et al. *Science Advances*, 8, 24 (2022)) proposed the concealing scheme by switching all the devices to low resistance states, but their approach cannot perform reconfiguration. John et al. (John, R. A. et al. *Nature Communications*, 12, 3681 (2021)) compared two neighboring cells to produce a PUF bit. Their PUF approach was already reported elsewhere, but the halide perovskite material was considered novel. Compared to these approaches, our PUF methodology allows for both reconfigurability and concealability which has not been reported yet. The proposed device bears another novelty that allows implementation in a passive crossbar array, which none of the prior works showed. Furthermore, demonstrating the logic operation capability is another breakthrough in this field because no previous devices have demonstrated security and computing capabilities in a single device. This is because the two applications require conflicting device operations. Most importantly, this work is not solely based on PUF but demonstrates a single system that performs PUF generation, logic operation, and encryption/decryption.

Specific changes: we added texts (on page 10) highlighting why the various functions relating to PUF generation are essential and how many prior efforts have tried to integrate them into a single component (and failed).

For (2), the authors agree that compared to the state-of-the-art Boolean logic, their demonstration remains primitive. I am all for primitive device concepts if they offer future promise, but in this case there is not even a Nikonov style benchmarking effort that would show how this primitive technology would scale.

The authors argue that while this is not competitive against CMOS (in theory or in practice), they are competitive against all other memristors approaches.

I am sorry but this is a very poor way of arguing why this work which brags "flexibility" of Boolean logic should be of broad interest to the community. Yes, these results may be better than all prior memristor proposals for Boolean logic, but that still doesn't change the fact that Boolean logic with memristors lags far behind the state-of-the-art and may sadly never materialize.

I cannot find such a line of reasoning convincing.

Answer: First, we want to stress again that this is the first demonstration of five different functions in a single device material (reconfigurability, concealability, nonvolatility, logic capability and scalability in a passive crossbar array configuration). We believe that we provided sufficient references and discussion in the revision to justify the need for these functions within a single component.

Second, we provided information on scalability in response to Referee 3, which is included in Supplementary Note 2, Supplementary Figs. 4-5.

Third, we cited Nikonov's work in the revision (in the Discussion section), and also compared our performance to CMOS computers. In fact, we use Nikonov's assertion that post-CMOS devices must be researched for logic operations to justify logic operations in our devices. In Nikonov's recent work, they only compared spintronic logic without direct comparison with CMOS. In this work, we compared a full adder scheme with memristor-based in-memory computing concepts regarding the number of required devices, operation steps, and energy cost. Then, we also provided energy and

latency comparisons of XOR encryption, which includes PUF generation and logic operation, between our approach and the CMOS-based computer.

Specific changes: we added these arguments in the Discussion section (Page 17) and the section on PUFs (Page 10).

(3) Finally, the authors respond to my criticism of their random number generation with some projections. Here, too, unfortunately, one must compare an actual chip from Intel, to the author's projections. I have no doubt these projections were prepared earnestly and with reasonable assumptions, it seems unfair to compare them to an actual demonstration.

Once again: I am not against primitive demonstrations that exhibit important, potentially ground-breaking phenomena if scaled. In this case, however, I do not see the novelty or the promising avenues this research opens up for it to be featured on the pages of Nature Communications.

Answer: We appreciate the reviewer for the careful comment. Nonetheless, we cautiously indicate that comparing projections to an actual demonstration is unfair. We strongly feel that scientific research needs such projections to guide the down-selection of the most promising research primitives for technology development. This approach is standard across electronics research (e.g., Cai, Kumar et al., *Nature Electronics*, 3, 409 (2020)).

As for the relevance of this work to the journal, we believe we provided sufficient references of papers published by *Nature Communications* with a similar level of novelty, which nonetheless represents the best of research.

Specific changes: We added some of these arguments in the Discussion section (Page 17).